# Mixed Hierarchical Oracle and Multi-Agent Benchmark in Two-player Zero-sum Games

## Abstract

Self-play methods have achieved remarkable success in two-player zero-sum games, attaining superhuman performance in many complex game domains. Parallelizing learners is a feasible approach to handling large-scale games. However, parallelizing learners often leads to suboptimal exploitation of computational resources, resulting in inefficiencies. In this study, we introduce the Mixed Hierarchical Oracle (MHO), designed to enhance computational efficiency and performance in large-scale two-player zero-sum games. MHO enables the parallelization of reinforcement learning tasks through a hierarchical pipeline that balances exploration and exploitation across oracle levels. It also avoids cold-start issues by using a "model soup" initialization strategy. Additionally, we present MiniStar, an open-source environment focused on small-scale combat scenarios, developed to facilitate research in self-play algorithms. Through extensive experiments on matrix games and the MiniStar environment, we demonstrate that MHO outperforms existing methods in terms of computational efficiency and performance.

## 1 Introduction

Two-player zero-sum games, often modeled as competitive interactions in sports and e-sports, present a rich area of study Ye et al. (2020a); Silver et al. (2017); Vinyals et al. (2019); Peng et al. (2017); Ye et al. (2020b). Advancements in artificial intelligence and self-play algorithms have led to superhuman performances in complex tasks across various games, including MOBA games Berner et al. (2019); Ye et al. (2020a;b); Wei et al. (2022), Go Silver et al. (2017), and StarCraft Peng et al. (2017); Vinyals et al. (2019). Despite the potential for a wide range of applications, self-play algorithms have limitations such as possible convergence to suboptimal strategies and high computational resource demands Yang et al. (2021).

Traditional self-play algorithms McMahan et al. (2003); Heinrich & Silver (2016); Hernandez et al. (2019) improve an intelligent agent's strategy by having it repeatedly play against itself, allowing the agent to explore a variety of strategies and enhance its decision-making capabilities without external training data. However, in complex game environments, these methods may struggle to find optimal strategies due to the vastness of the strategy space. To address this challenge, the Policy Space Response Oracle (PSRO) algorithm Lanctot et al. (2017) extends the Double Oracle (DO) algorithm McMahan et al. (2003) to large-scale games by using reinforcement learning (RL) to approximate best responses. Empirical Game-Theoretic Analysis (EGTA) Wellman (2006); Wellman et al. (2024) is a framework for studying meta-strategies obtained through simulation in complex games. PSRO combines EGTA with RL and introduces a Meta-Strategy Solver (MSS) to assist in selecting adversarial strategies, thereby improving strategy selection in self-play and ensuring convergence towards an approximate Nash equilibrium.

To effectively handle large-scale game scenarios, parallelization methods are often employed to solve best-response problems Lanctot et al. (2017); McAleer et al. (2020). Deep Cognitive Hierarchies (DCH) Lanctot et al. (2017) and Pipeline PSRO (P2SRO) McAleer et al. (2020) exploit the iterative nature of the Policy Space Response Oracle (PSRO) algorithm to parallelize oracle computations. However, their approach to oracle parallelism allocates computational resources almost equally among all oracles, which can lead to suboptimal computational efficiency. Moreover, in P2SRO, when active policies generate data by playing against each other, the data produced by lower-level active policies is not utilized for training, resulting in data inefficiency. Additionally,

PSRO relies on EGTA to compute the meta-game, which incurs higher computational overhead as the number of training iterations increases and the environment becomes more complex.

To address these issues, we introduce the Mixed Hierarchical Oracle (MHO), an abstraction and improvement based on the P2SRO method. MHO removes the dependence on EGTA to reduce extra overhead during training and improve training efficiency. MHO incorporates three key approaches: parallelizing oracles at different levels to enlarge the solver's capacity; utilizing samples generated when high-level oracles compete against low-level oracles for training the corresponding oracles; and initializing newly added oracles using a "model soup" strategy to avoid cold starts when competing against low-level policier. Different levels of oracles have distinct exploration factors, with high-level policies favoring exploration and low-level policies favoring exploitation. Since MHO is an abstraction of P2SRO, it can also be applied to PSRO to improve its performance.

We also introduce a mini-environment called MiniStar, developed from StarCraft II (SC2)Vinyals et al. (2019) and SMACv2 Ellis et al. (2024), tailored to focus on small-scale combat scenarios and self-play research. By narrowing the scope to tactical engagements rather than full-game strategies, we reduce the complexity associated with long-time-series decision-making. This simplification allows us to concentrate on the core aspects of self-play algorithms without the prerequisite of extensive reinforcement learning optimization.

To summarize, in this paper we provide the following contributions:

- We propose MHO, a self-play algorithm that enhances computational efficiency and performance in large-scale gaming scenarios by avoiding cold starts and balancing exploration and exploitation across oracle levels.
- We introduce MiniStar, small-scale combat scenarios specifically designed for self-play research.
- We demonstrate the effectiveness of MHO through extensive experiments on matrix games and the MiniStar environment.

## 2 RELATED WORKS

**Self-play Methods** In self-play methods, agents are trained by repeatedly playing against their latest versions. Fictitious Self-Play (FSP) Heinrich et al. (2015) enables agents to play against their past selves to learn optimal strategies. Neural Fictitious Self-Play Heinrich & Silver (2016) is a modern variant that combines FSP with deep learning techniques, using neural networks to approximate the best response (BR). Preferred Fictitious Self-Play (PFSP) Vinyals et al. (2019) utilizes a preference function to assign higher selection probabilities to higher-priority agents. The Double Oracle (DO) McMahan et al. (2003) method approximates Nash Equilibria in large-scale zero-sum games by iteratively creating and solving a series of sub-games with a restricted set of pure strategies. Policy Space Response Oracle (PSRO) Lanctot et al. (2017) is a generalization of DO, using reinforcement learning as an oracle to enable decision-making in complex gaming environments. It introduces the concept of a Meta-Strategy Solver (MSS) to assist in the selection of adversarial strategies, which guarantees convergence to an approximate Nash equilibrium. Pipeline PSRO (P2SRO) McAleer et al. (2020) realizes the parallelization of PSRO while simultaneously guaranteeing convergence. Online Double Oracle (ODO) Dinh et al. (2022) combines no-regret analysis from online learning with the DO approach to improve the speed of convergence to a Nash equilibrium and the average payoff. Anytime PSRO McAleer et al. (2022b) and Self-Play PSRO McAleer et al. (2022a) aim to incorporate more difficult-to-exploit strategies into the strategy population, thus facilitating faster convergence. These variants further enhance the performance and applicability of the self-play method by introducing new strategy evaluation mechanisms and optimizing the strategy selection process.

**Simulation Environment** Compared to traditional board and card games, simulation environments are typically characterized by real-time operations, long periods, and higher complexity of environmental state transitions, such as StarCraft II (SC2) Vinyals et al. (2019), Google Research Football (GRF) Kurach et al. (2020), and Multiplayer Online Battle Arena (MOBA) games like Dota 2 Berner et al. (2019) and Honor of Kings Ye et al. (2020a;b); Wei et al. (2022). These environments present agents with real-time, partially observable settings requiring continuous decision-making over extended time horizons. Agents must handle large, continuous action spaces and deal with uncertain-

ties introduced by dynamic opponents and environments. The complexity and high dimensionality of these environments necessitate extensive reinforcement learning and engineering optimization before effective self-play can be conducted. For instance, AlphaStar Vinyals et al. (2019) combines reinforcement learning, self-play, and imitation learning to achieve master-level performance using vast computational resources. OpenAI Five Berner et al. (2019) demonstrated that self-play could be scaled to achieve superhuman performance in Dota 2 by training agents in a massively parallel framework. Similarly, efforts in Honor of Kings have focused on developing AI that can operate in complex team-based settings, emphasizing cooperation and coordination among agents Ye et al. (2020a). Although there are success stories in self-play research, they all heavily rely on mature engineering practices, which undoubtedly raises the bar for academic research. GRF-based studies such as TiZero Lin et al. (2023), TiKick Huang et al. (2021), and others require the same upfront work of pre-training and model optimization to address training difficulties caused by cold starts before proceeding to self-play.

## 3 PRELIMINARIES

### 3.1 TWO-PLAYER NORMAL-FORM GAMES

A two-player normal-form game Fudenberg & Tirole (1991) is characterized by the tuple $(\mathcal{A}, \mathbf{U})$, where $\mathcal{A} = (\mathcal{A}_1, \mathcal{A}_2)$ represents the action sets for each player $i \in \{1, 2\}$, and $\mathbf{U} = (u_1, u_2)$ denotes their respective utility functions. Formally, for each player $i$, the utility function $u_i : \mathcal{A}_1 \times \mathcal{A}_2 \to \mathbb{R}$ assigns a real-valued payoff to every possible action pair.

Players aim to maximize their expected utility by choosing a mixed strategy $\pi_i \in \Delta(\mathcal{A}_i)$, where $\Delta(\mathcal{A}i)$ denotes the set of probability distributions over $\mathcal{A}i$. For notational convenience, we denote the opponent of player $i$ as $-i$. The best response to an opponent's mixed strategy $\pi_{-i}$ is the mixed strategy $(\mathrm{BR}(\pi_{-i}))$ that maximizes player $i$'s utility:

$$\mathrm{BR}(\pi_{-i}) = \arg \max_{\pi_i} G_i(\pi_i, \pi_{-i}), \tag{1}$$

where $G_i(\pi_i, \pi_{-i}) = \mathbb{E}_{a_i \sim \pi_i, a-i \sim \pi_{-i}}[u_i(a_i, a_{-i})]$ denotes the expected utility for player $i$ given the mixed strategies $\pi_i$ and $\pi_{-i}$.

### 3.2 META STRATEGY

The concept of a meta-game extends the game to a higher level of abstraction by considering a population of policies $\Pi_i = \pi_i^1, \pi_i^2, \ldots$ for each player $i$. In this context, choosing an action corresponds to selecting a specific policy from the set $\Pi_i$. The interactions within this expanded policy space are captured by the payoff matrix $M_{\Pi_i, \Pi_{-i}}$, where $M_{\Pi_i, \Pi_{-i}}[j, k] = G_i(\pi_i^j, \pi_{-i}^k)$. Here, $G_i(\pi_i^j, \pi_{-i}^k)$ denotes the expected utility for player $i$ when using policy $\pi_i^j$ against the opponent's policy $\pi_{-i}^k$.

In the meta-game, a meta-strategy $\sigma_i$ represents a mixed strategy over the policy set $\Pi_i$, assigning probabilities to each policy in the set. Meta-games are often open-ended because an infinite number of mixed strategies can be constructed from the available policies.

In self-play methods, each player $i$ maintains a set of strategies $\Pi_i$ for themselves and observes the opponent's strategy set $\Pi_{-i}$. This framework allows for the construction of meta-strategies $\sigma_i$ to elucidate the dynamics between players' strategies. The meta-strategy $\sigma_i$ for player $i$ is derived from various solvers such as Nash Equilibrium (NE), Fictitious Play (FP) Brown (1951), or Preferred Fictitious Self-Play (PFSP).

When a new policy $\pi_i'$ is introduced, the framework recalculates the best response, often referred to as the *Oracle*. If the Oracle is determined through reinforcement learning, the best response is represented as:

$$\mathrm{Oracle}(\sigma_{-i}) = \arg \max_{\pi_i'} \sum_j \sigma_{-i}^j E_{\pi_i', \pi_{-i}^j}[R], \tag{2}$$

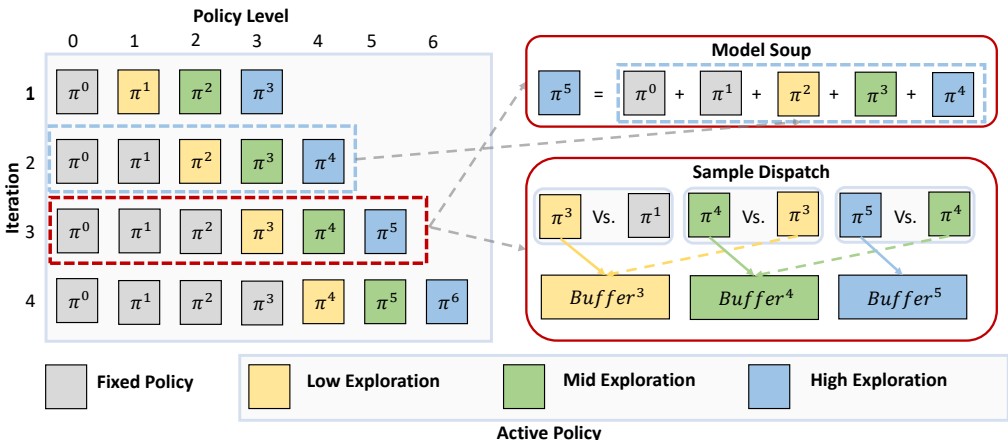

Figure 1: The overall framework diagram of MHO. The policy in MHO consists of Fixed Policy which is fixed and active policy which is being trained. Active policy is a set of parallel hierarchical policies. Higher-level policies are more exploratory in training and lower-level policies are more exploitative. After the lowest level policy (yellow in the figure) completes training it becomes a fixed policy. A new active policy is added as the highest-level policy (blue in the figure) and initialized by the lower-level policy using the model-SOUPING method. After the higher-level policy finishes fighting against the lower-level active policy, the samples are learned by the respective level active policy, instead of discarding the samples of the lower-level active policy McAleer et al. (2020).

where $R$ denotes the reinforcement learning reward, typically configured in a zero-sum setting, and $E_{\pi_i', \pi_{-i}^j}[R]$ represents the expected reward when player $i$ uses policy $\pi_i'$ against the opponent's policy $\pi_{-i}^j$.

To quantify how far the joint strategy profile $\sigma = (\sigma_1, \sigma_2)$ is from a Nash equilibrium, we use exploitability, measured by NashConv Lanctot et al. (2017):

$$\text{Expl}(\sigma) = \sum_{i=1}^{2} \left[ G_i(\text{BR}(\sigma_{-i}), \sigma_{-i}) - G_i(\sigma_i, \sigma_{-i}) \right], \tag{3}$$

where $\text{BR}(\sigma_{-i})$ denotes the best response to the opponent's meta-strategy $\sigma_{-i}$. When the exploitability reaches zero, the joint mixed strategy $\sigma$ corresponds to a Nash equilibrium.

## 4 METHODOLOGY

In this section, we introduce the Mixed Hierarchical Oracle (MHO) methods that significantly improve the training and performance of reinforcement learning agents in self-play algorithms, especially in complex environments. MHO comprises three key components: Parallelized Oracle, Model Souping, and Hierarchical Exploration. These components aim to address the computational efficiency of the policy learning process from different perspectives—sample utilization, cold-start issues, and exploration mechanisms—and are ultimately integrated to work synergistically. The detailed framework is illustrated in Figure 1.

### 4.1 PARALLELIZED ORACLE

We propose a parallelized active policy sampling method that enhances the interaction between active policies during training. In the parallelized learning of P2SRO, each active policy during training is allocated an equal portion of the total computational resources, and data are collected independently for training, which reduces training efficiency. the P2SRO method was modified to improve the data utilization after sampling, where each high-level active policy interacts with

a lower-level active policy against post-sampling, and the resulting data samples are distributed to the corresponding active policies for learning. Each active policy interacts with the environment and other policies based on the meta-strategy $\sigma_{-i}$. the meta-strategy $\sigma_{-i}$ is calculated using the FSP over the set of lower-level policies, which avoids the higher computational cost associated with computing the meta-Nash equilibrium via EGTA. This approach balances computational efficiency and strategic learning, making it suitable for large-scale complex environments

In our proposed Parallelized Oracle approach, we denote $w_{\text{active},j}$ as the total sampling probability of the $j$-th policy interacting with all other active policies except itself. Let $m(m < j)$ be the number of active policies and $n$ be the total number of environments or available samples. For the $k$-th active policy, the cumulative increase in the number of samples from other active policies $\Delta S_k$ is given by:

$$\Delta S_k = \frac{n}{m} \times w_{\text{active},j-m+k} \tag{4}$$

The total increase number of samples $\Delta S_{\text{total}}$ is then adjusted to:

$$\Delta S_{\text{total}} = \frac{n}{m} \times \sum_{k=1}^{m} w_{\text{active},j-m+k} \tag{5}$$

During parallel training, each active policy further utilizes the data generated by the high-level policy while training with the fixed policy. The method effectively scales with the number of active policies to optimize computational resources.

### 4.2 MODEL SOUPING

At the model initialization stage of each new training round, if the model is initialized from scratch, agents may struggle to learn effective policies in the later stages of self-play as opponents become stronger. To avoid this cold-start problem, we employ a model fusion method, known as "model souping," to achieve a warm start.

Specifically, after each round of training, a new top-level active policy is obtained by parameter fusion of the lower-level active policies and the fixed policies. This fusion shares the knowledge learned among different active policies, enhancing data utilization and accelerating learning in subsequent training rounds.

In the context of model souping, we employ a meta-strategy as a weighted combination of the model parameters from the lower-level policies. Mathematically, for the set of policies $\Pi_i = \{\pi_i^1, \pi_i^2, \ldots, \pi_i^j\}$, and $\theta_{\pi_i^j}$ denotes the parameters of the $j$-th policy, under the meta-strategy $\sigma_i$, the new policy $\theta_{\pi_i^{j+1}}$ is computed as:

$$\theta_{\pi_i^{j+1}} = \sum_{k=1}^{j} \sigma_i^k \cdot \theta_{\pi_i^k} \tag{6}$$

Model Souping mitigates the computational fragmentation caused by Parallelized Oracle by fusing model parameters across different learners, effectively recombining the split computational resources. This fusion enhances data utilization by sharing knowledge within the policy pool, overcoming the inefficiency of independent learning in parallelized settings

### 4.3 HIERARCHICAL EXPLORATION

In self-play algorithms, it is often necessary to truncate the approximate best-response operator at each iteration, which can lead to suboptimal training outcomes. To migrate this issue, we introduce a hierarchical exploration mechanism within the Parallelized Oracle framework, where different exploration factors are assigned to different tiers of the active policy pool.

Specifically, the highest-level active policies are more inclined to explore during training after initialization and gradually shift towards exploitation as training progresses. To achieve this, we incorporate an entropy regularization term in the computation of the optimal response. The entropy term

encourages exploration by penalizing deterministic behavior in the policy, thus promoting stochasticity during training.

Let $H(\pi)$ represent the entropy of a policy $\pi$. The objective function for Equation 2 is modified as follows:

$$\text{Oracle}(\sigma_{-i}) = \arg \max_{\pi_i'} \sum_j \sigma_{-i}^j E_{\pi_i', \pi_{-i}^j}[R] - \lambda_k H(\pi_i') \tag{7}$$

Here, $\lambda_k$ is a hyperparameter that controls the strength of the entropy regularization. As training progresses, $\lambda_k$ decreases in tandem with the shift of policies from high-active to low-active, with the highest-level policy having the largest value of $\lambda_k$ and the lowest-level policy having the smallest. This synchronized reduction of $\lambda_k$ ensures that exploration is encouraged early in the training, while policies progressively focus more on exploitation as they transition towards lower activity levels. This mechanism effectively maintains a balance between exploration and exploitation across different policy tiers, ensuring that agents explore sufficiently in the early stages, while refining and exploiting learned strategies in the later stages.

## 5 EXPERIMENTS

### 5.1 REAL-WORLD META-GAME

AlphaStar888 Czarnecki et al. (2020) is an empirical game derived from the solution process of StarCraft II Vinyals et al. (2019), featuring a payoff table involving 888 reinforcement learning strategies. It can be viewed as a zero-sum symmetric two-player game with only one state. In this state, there are 888 legal actions, and any mixed strategy corresponds to a discrete probability distribution over these actions. Due to the size of the $888 \times 888$ payoff matrix, the computational time required for strategy optimization can significantly vary among algorithms, making it suitable for demonstrating differences in time overhead.

In this experiment, we additionally included the REFINED algorithm for comparison to enrich the experimental results. The experimental results are shown in Figure 2. During the experiments, we recorded the exploitability Lanctot et al. (2017) of each algorithm's current policy set and the computation time each algorithm spent on performing policy optimization. We plotted the exploitability against the computation time, with computation time on the horizontal axis and exploitability on the vertical axis.

The experimental results in Figure 2a show that the exploitability of the MHO algorithm decreases significantly and improves even further when combined with the PSRO algorithm. This improvement is mainly due to the use of Nash Equilibrium (NE) as the weighting mechanism in the model soup within PSRO. Although the MHO algorithm slightly increases the total computation time for strategy optimization, it plays a crucial role in enhancing the overall performance of the algorithm. Additionally, comparisons of SP vs. MHO, PSRO vs. P2SRO, and PSRO vs. PSRO with MHO indicate that due to the presence of the Parallelized Oracle, computational resources are partitioned, which leads to a decrease in the speed of convergence in the early stages. However, in the long term, MHO demonstrates more robust performance.

We conducted two ablation experiments to verify the performance of Parallelized Oracle, Model Souping, and Hierarchical Exploration in MHO using Self-Play and the PSRO. The experimental results are shown in Figures 2b. From these results, we observe that Model Souping has the greatest impact on improving strategy performance. The impact of Hierarchical Exploration varies across different algorithms due to differing hyperparameters, leading to different performance outcomes. Parallelized Oracle enhances the performance of strategy optimization by increasing data utilization.

### 5.2 MINISTAR

The MiniStar environment is a simplified version of StarCraft designed specifically for skirmish scenarios and self-play research. By focusing on localized battle control rather than the full spectrum of StarCraft gameplay—which includes resource management, mission planning, and large-scale battle control—MiniStar allows agents to concentrate on the micro-level manipulation of decision-making actions. This targeted approach reduces the complexity of the environment, enabling more

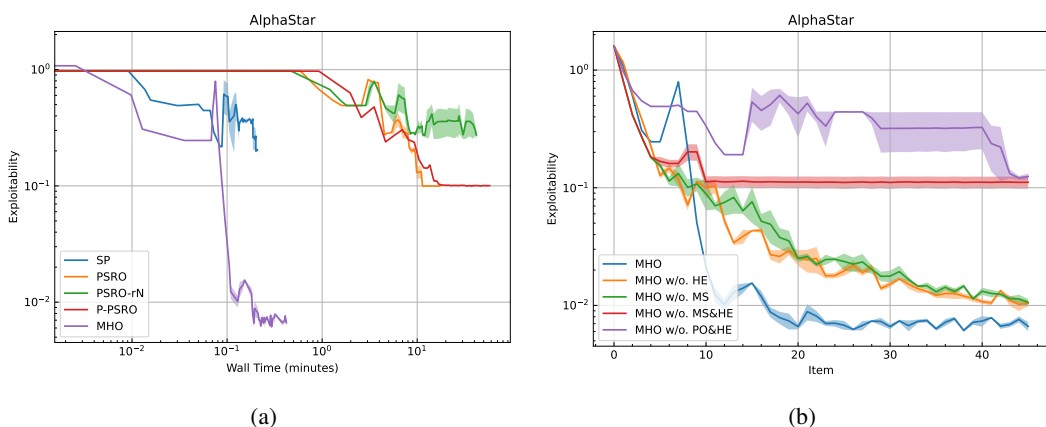

(a)            (b)

Figure 2: The experiments in AlphaStar888. (a) Main experimental results comparing different algorithms, with exploitability plotted against wall-clock time (in minutes). (b) Ablation experiment on AlphaStar888 comparing the performance of MHO and SP, with exploitability plotted against training steps.

efficient development of zero-sum game algorithms in focused combat situations. In the SMAC and SMACv2 environments, agents control one faction while the opposing faction is managed by a built-in bot, and there is no support for agents to control both factions. In contrast, MiniStar extends SMACv2 by allowing agents to control both factions simultaneously in a self-play setting, eliminating the need for built-in bots.

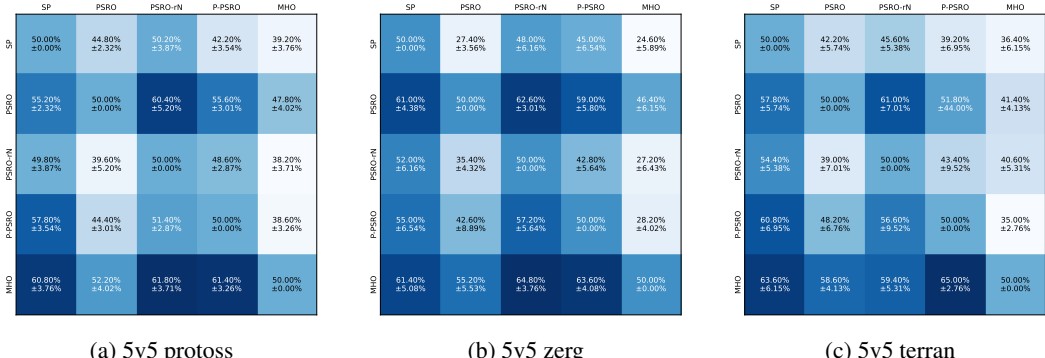

(a) 5v5 protoss        (b) 5v5 zerg        (c) 5v5 terran

Figure 3: The Meta-game matrix between different algorithms

We tested each of the three races in that experiment, all in 5v5 matchmaking mode. For Zerg, they are zergling, hydralisk, and baneling; for Terran, they are marine, marauder, and medivac; and for Protoss, they are stalkers, zealots, and colossi. the three racial unit weights relative to a fixed unit order are [0.45,0.45,0.1], and birth locations are randomized for the Surround and Reflect scheme. All algorithms are trained on 20M steps. The experimental results presented in Figure 3 show that both the MHO algorithm and the PSRO method integrated with MHO(PSRO w. MHO) perform exceptionally well.

## 6 CONCLUSION

In this paper, we introduced the Mixed Hierarchical Oracle (MHO) algorithm from the perspective of parallelized reinforcement learning solvers to enhance computational efficiency and policy performance. MHO addresses issues of low computational efficiency and cold-start problems caused by parallelism by enhancing interaction between different parallelized modules. The core of the algorithm lies in its parallelization strategy, which improves the effectiveness of policy learning

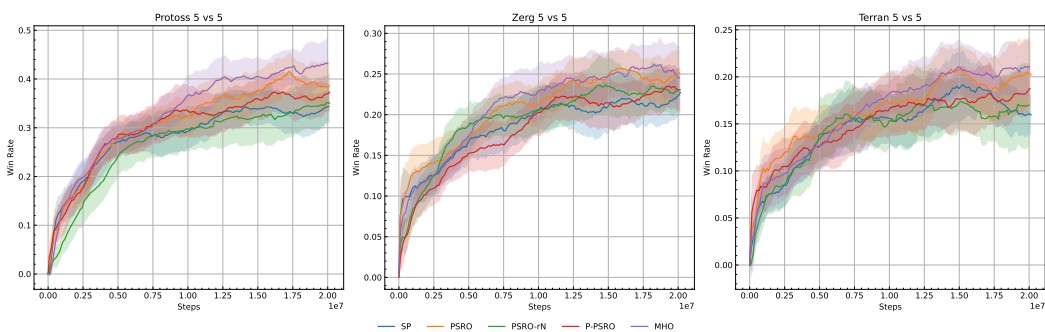

Figure 4: Winning curves of different algorithms against built-in AI during training.

through increased inter-module communication. Furthermore, recognizing the lack of focused simulation scenarios in current two-player zero-sum game environments for game research, we developed the MiniStar environment. MiniStar reduces the engineering complexity associated with zero-sum game research, providing a lighter and more flexible research scenario for the community. Using the MHO algorithm, we trained a series of policies on MiniStar, demonstrating that even algorithms like PSRO, which require Empirical Game-Theoretic Analysis (EGTA), can be trained with acceptable computational overhead.

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
