# OpenReview forum: "Mixed Hierarchical Oracle and Multi-Agent Benchmark in Two-player Zero-sum Games"
_ICLR.cc/2025/Conference — ICLR 2025 Conference Withdrawn Submission_

### Official Review · Reviewer_V53Q · 2024-11-02

**Soundness:** 3
**Presentation:** 2
**Contribution:** 3
**Rating:** 5
**Confidence:** 2

**Summary:**

This paper proposes Mixed Hierarchical Oracle (MHO): an improved version of Pipeline Policy Space Response Oracle (P2SRO) for self-play learning of strategies in two-player zero-sum games. It primarily aims to improve performance by improving computational efficiency in large-scale parallel training, avoiding cold starts in policies, and adjusting which policies perform how much exploration/exploitation.

**Strengths:**

- I think the algorithms are sound and sensible (though I do have some concerns about my own ability to judge this well).
- Empirical results look good.

**Weaknesses:**

My primary concern with this paper is that my impression is that it builds quite heavily on top of P2SRO, but provides very little background information on P2SRO itself. There is a good Preliminaries section, and the things it explains are explained well, but I feel that there is still a bit of a gap between where the Preliminaries section ends, and understanding P2SRO. As someone who is not quite familiar with P2SRO, I find that this makes it very difficult for me to appreciate / understand exactly what your changes are doing (or, more precisely, I think I do not fully grasp how the things that you changed relative to P2SRO even worked to begin with).

4.2 describes doing "parameter fusion", but there is no explanation of what this is. I can guess it means something like averaging parameters, but I shouldn't have to guess. Please provide precise and exact technical details on the techniques proposed/used.

Section 5.1 suddenly mentions the inclusion of a "REFINED" algorithm, but I have no idea what this is or means.

Another concern of mine is that the introduction of MiniStar as a benchmark/framework is described as a secondary contribution of the paper, but very little detail on this framework is provided. The paper basically only describes that this is meant to focus on smaller combat scenarios, as opposed to the full game of e.g. StarCraft 2. My understanding is that SMACv2 also already does this though. How is MiniStar different?

The legends, axis labels and tick labels of Figure 2 are too small.

**Questions:**

1) How does MiniStar differ from SMACv2?

---

### Official Review · Reviewer_4Vzf · 2024-11-03

**Soundness:** 1
**Presentation:** 2
**Contribution:** 1
**Rating:** 3
**Confidence:** 4

**Summary:**

This paper introduces the Mixed Hierarchical Oracle (MHO), a novel self-play algorithm designed to enhance computational efficiency and performance in large-scale two-player zero-sum games. MHO builds upon existing methods like Policy Space Response Oracle (PSRO) and Pipeline PSRO (P2SRO) by parallelizing reinforcement learning tasks through a hierarchical pipeline that balances exploration and exploitation across oracle levels. It addresses issues such as suboptimal resource allocation, data wastage, and cold-start problems by utilizing a "model soup" initialization strategy and removing dependence on Empirical Game Theoretic Analysis (EGTA). The paper also introduces MiniStar, an open-source environment focused on small-scale combat scenarios to facilitate research in self-play algorithms. Through experiments on matrix games and the MiniStar environment, the study demonstrates that MHO outperforms existing methods in terms of computational efficiency and performance.

**Strengths:**

1. Clear introduction and helpful preliminaries: The authors have done well in setting up the context and providing necessary background information, which aids reader comprehension.

2. Novel algorithm: The paper introduces the Mixed Hierarchical Oracle (MHO), a new self-play algorithm that addresses multiple issues in existing methods. MHO combines three key components - Parallelized Oracle, Model Souping, and Hierarchical Exploration - to enhance computational efficiency and performance in large-scale two-player zero-sum games.

**Weaknesses:**

1. Unclear methodology section: The paper should be revised to provide clearer explanations of their proposed Mixed Hierarchical Oracle (MHO) approach. They should further break down each component (Parallelized Oracle, Model Souping, and Hierarchical Exploration) and explain how they work together to address the stated issues. For instance, it is not explicitly stated how the model souping combines models algorithmically or mathematically.

2. Insufficient experiments section: The paper should provide a clear rationale for each experiment, explicitly linking them to the paper's claims about addressing suboptimal resource allocation, data wastage, and cold-start problems. Figure explanations need improvement. Each figure should have a detailed caption and be thoroughly discussed in the text.

3. Experimental design and presentation issues: In Figure 2, the comparison of wall clock time for exploitability is inconsistent. The paper should have experiments that either run all methods for the same duration or justify the varying durations. The use of only tabular methods (AlphaStar888) limits the generalizability of the results. The paper should either include non-tabular experiments or provide a strong justification for focusing solely on tabular cases. More details about the AlphaStar888 environment should be provided, along with a defense of its usage as the primary testbed.

4. Figure quality and clarity: Figure 3 needs significant improvement. A legend explaining the payoff matrix is essential, and the paper should clarify whether higher or lower values are better. The matrix should be labeled to indicate which method corresponds to the row or column player.

5. Lack of novelty of MiniStar environment: The novelty of the MiniStar environment is questionable, as SMAC and SMACv2 (cited in the paper) already provide smaller-scale combat scenarios for multi-agent reinforcement learning research. The paper fails to clearly articulate how MiniStar offers unique advantages or capabilities beyond what these existing environments already provide.

**Questions:**

Suggestions for improvement:
1. Revise the methodology section to provide clearer explanations of MHO components and their integration.
2. Expand the experiments section:
- Add clear rationales for each experiment, linking them to the paper's claims.
- Improve figure explanations and captions.
- Consider adding non-tabular experiments or justify the focus on tabular cases.
- Provide more details about the AlphaStar888 environment.

3.  Redesign Figure 3 for better clarity and professional presentation.
5. There are plenty of papers left in the paper (over 3.5) to add additional content. Consider adding a discussion section to address limitations and potential future work.


Please see weaknesses for more details.

---

### Official Review · Reviewer_BRLq · 2024-11-03

**Soundness:** 2
**Presentation:** 2
**Contribution:** 2
**Rating:** 3
**Confidence:** 4

**Summary:**

The main contribution of the paper is MHO, a novel, parallelized, population-based algorithm for computing good strategies in 2-player, zero-sum games. The methodology is similar to Pipeline PSRO (P2SRO) [0], which is a sound method for parallelizing PSRO.

One of the main novelties of MHO is that when two actively training (i.e. not yet frozen) policies play each other, the experience generated by the play will be used to train both policies, instead of just the higher-level one. The other two contributions are Model Souping, wherein new policies are warm-started by initializing their weights by parameter fusion of existing policies, and Heirarchical Exploration, wherein each new policy will play with high exploration early on and switch to high exploitation later.

The other main contribution is a new simplified version of Starcraft designed for "skirmish scenarios and self-play research".

The paper presents experiments on both the AlphaStar 888x888 matrix game and the MiniStar environment, using their MHO, PSRO, P2SRO, and other baselines.

[0]: McAleer, Stephen, John Lanier, Roy Fox, and Pierre Baldi. 2020. “Pipeline PSRO: A Scalable Approach for Finding Approximate Nash Equilibria in Large Games.” ArXiv.org. 2020. https://arxiv.org/abs/2006.08555.

‌

**Strengths:**

The contributions of the paper are all reasonable potential improvements to population-based algorithms for 2P0S games. In particular, it's a legitimate research question to investigate what happens empirically if lower-level policies can train against higher-level policies. Model-souping and Heirarchical Exploration are also reasonable ideas to try. The new Ministar environment also seems like a valuable environment for researchers.

**Weaknesses:**

Unfortunately, the paper has several flaws that prevent me from recommending it for acceptance.

Several parts of the paper, including the core algorithm itself, are not clearly explained.
1. The algorithm is simply never outlined in the paper -- neither in natural language nor in pseudocode.
2. e.g. Section 4.1 (Parallelized Oracle): how is the meta-strategy calculated? What is the overall structure of the algorithm? Is it like PSRO?
3. e.g. Section 4.2 (Model Souping): No details are given for model souping, only a very high-level description. If this is an existing, off-the-shelf method, it should be cited.
4. e.g. Section 4.3 (Hierarchical Exploration): No explanation is given for how exploration and exploitation are controlled.
5. It's not clear what the solution concept is. Although it is alluded to, it is never clearly stated that the goal is to find an approximate Nash equilibrium strategy (e.g. one with low exploitability).
6. Experiments are conducted with "MHO" and with "MHO with PSRO", but it is never explained what the difference is.

There are some incorrect or inscrutable statements in the exposition.
1. Introduction: "these methods may struggle to find optimal strategies due to the vastness of the strategy space.", but also because the methods may not be sound: they may cycle or exhibit chaotic behavior even in simple strategy spaces; Nash equilibria may not even be fixed points of the algorithms.
2. Introduction: "Additionally, PSRO requires the use of EGTA to compute the meta-game, incurring higher computational overhead as training iterations increase and the environment becomes more complex" and "MHO removes the dependence on EGTA to reduce extra overhead and improve training efficiency" -- I don't understand these sentence at all. What does "the use of EGTA" mean?
3. Section 3.2: "Meta-games are often open-ended because an infinite number of mixed strategies can be constructed from the available policies." -- what does "open-ended" mean? And "This framework allows for the construction of meta-strategies to elucidate the dynamics between players' strategies" -- what does this mean?
4. Section 3.2: What does it mean that the meta-strategy is derived from various solvers such as "Nash equilibrium, fictitious play [...]"? I would say that the meta-strategy can be things like the Nash equilibrium, or uniform random, and that the Nash equilibrium can be computed via an LP or any other method.

The experiments are paltry and not presented well.
1. The details of the AlphaStar888 experiments are not given. How are the oracles trained? How is the meta-strategy computed? Additionally, I question that a matrix game is a good environment to demonstrate the algorithm, since computing an oracle could be done almost instantly (by picking the row/column with the highest expected payoff).
2. Results are given in wall-time (Figure 2a) and training steps (Figure 2b) [although the x-axis is labelled "item" in 2b]. It's not clear what the training steps refer to (outer-loop iterations? raw experience?). Figure 2a should also be given with iterations or training steps as the x-axis.
3. Hyperparameters for the baselines are not given. Hyperparameters for the main algorithm are not given (nor are the possible hyperparameters even described).

Soundness of the algorithm and a comparison of soundness to PSRO, P2SRO, DCH, etc. are not included. This is important because the missing context from the paper is that presumably P2SRO does not train lower-level policies against higher-level policies because doing so does not guarantee convergence to a Nash equilibrium.

Citations aren't parenthesized. The paper should note that Self-Play PSRO is similar to MHO in that lower-level policies are trained with experience playing against higher-level policies.

**Questions:**

Questions listed above.

---

### Note · Authors · 2024-11-27

I have read and agree with the venue's withdrawal policy on behalf of myself and my co-authors.